# Dietary Sodium and Potassium Intake: Data from the Mexican National Health and Nutrition Survey 2016

**DOI:** 10.3390/nu14020281

**Published:** 2022-01-11

**Authors:** Jorge Vargas-Meza, Manuel A. Cervantes-Armenta, Ismael Campos-Nonato, Claudia Nieto, Joaquín Alejandro Marrón-Ponce, Simón Barquera, Mario Flores-Aldana, Sonia Rodríguez-Ramírez

**Affiliations:** Center for Nutrition and Health Research, National Institute of Public Health, Av. Universidad 655 Col Santa María Ahuacatitlán, Cuernavaca C.P. 62100, Morelos, Mexico; jorge.vargas@insp.mx (J.V.-M.); mcervantesarmenta@gmail.com (M.A.C.-A.); claudia.nieto@insp.mx (C.N.); joaquin.marron@hotmail.com (J.A.M.-P.); sbarquera@insp.mx (S.B.); mario.flores@insp.mx (M.F.-A.); scrodrig@insp.mx (S.R.-R.)

**Keywords:** sodium, potassium, salt, processed foods, ultra-processed foods, national survey, Mexico

## Abstract

Population studies have demonstrated an association between sodium and potassium intake and blood pressure levels and lipids. The aim of this study was to describe the dietary intake and contribution of sodium and potassium to the Mexican diet, and to describe its association with nutrition status and clinical characteristics. We analyzed a national survey with 4219 participants. Dietary information was obtained with a 24-h recall. Foods and beverages were classified according to level of processing. The mean intake (mg/d) of Na was 1512 in preschool children, 2844 in school-age children, 3743 in adolescents, and 3132 in adults. The mean intake (mg/d) of K was 1616 in preschool children, 2256 in school-age children, 2967 in adolescents, and 3401 in adults. Processed and ultra-processed foods (UPF) contribute 49% of Na intake in preschool children, 50% in school-age children, 47% in adolescents, and 39% in adults. Adults with high Na intake had lower serum concentrations of cholesterol, HDL-c, and LDL-c. A significant proportion of the Mexican population has a high intake of Na (64–82%) and low K (58–73%). Strategies to reduce Na and increase K intake need to reduce the possibility of having high BP and serum lipid disturbances.

## 1. Introduction

Sodium (Na) and Potassium (K) are essential minerals for human homeostasis because they help maintain osmotic balance [1]. In the plasma membrane of cells, Na and K move against concentration gradients through constant pumping of Na-K ATPase, regulating cell volume and maintaining homeostasis of tissues and organs [2].

Population studies have demonstrated an association between high dietary Na intake and high blood pressure (BP) [3] and metabolic disorders [4]. Conversely, a low Na intake can lower BP in people who are sensitive to salt, increase the vascular insulin resistance, and improve blood levels of cholesterol and triglycerides [5].

The Institute of Medicine (IOM) established a maximum tolerable intake level (UL) of Na of 2.3 g/day [6], while the World Health Organization (WHO) recommends not exceeding 2 g/day [7]. Both organizations recommend limiting Na intake because higher consumption is associated with adverse health effects [8]. In a cohort study conducted in Mexico, the mean Na intake was 3.5 g/day [9].

In Latin America, sodium intake has been identified as coming from different food sources. Argentina has shown that the highest sodium intake comes from ultra-processed foods (UPF) (65–70%), while in Brazil and Costa Rica the main source is common salt (74 and 60%, respectively) [10].

In Mexico, the purchase of UPF products increased during the last decades [11]; however, their dietary contribution to daily sodium intake is unknown. In a study that examined sodium levels of UPF for 14 countries in Latin America and the Caribbean, including Mexico, the foods with the highest sodium content were unprocessed and processed meats, ready-to-eat cereals, yogurt and milk-based drinks, seasonings, and salty snacks [12].

On the other hand, adult K intake was low (1.9 g/day) in a Mexico City cohort [9], compared to international recommendations (at least 3.5 g/day) [6,13]. Adequate K intake has been shown to have beneficial effects, such as lowering blood pressure and risk of kidney disease [5,14]. However, K is among the four most common deficient nutrients in the world diet [9,15].

The WHO highlights the importance of implementing a surveillance system to measure and evaluate the patterns of salt intake, as well as identify the main sources of Na intake in the population, since this can inform the development of programs aimed at reducing Na intake [16]. Furthermore, the effects of high sodium and low potassium intake on high BP [3], reduced renal function, and metabolic disorders are synergic, so the Na-K ratio may be a strong indicator for cardiovascular health outcomes [17,18]. The current evidence indicates that a ratio greater than 1.0 between Na and K results in an increase in BP [19].

There have been efforts in Mexico to measure Na and K intake [9,20]; however, these studies are not nationally representative and do not include all age groups. To date, there are no data about the national intake of Na and K or their food sources. Therefore, the aim of this study was to describe the dietary intake and main food sources of Na and K in participants of the Mexican National Health and Nutrition Survey 2016 (ENSANUT-2016), and to describe the association between Na and K intake with cardiovascular health outcomes.

## 2. Materials and Methods

### 2.1. Design and Study Population

This cross-sectional study used data from the ENSANUT-2016, which is a probabilistic survey at the national, regional, and state level with urban and rural strata. The data collection was between May and October of 2016. Details of the design, sampling size calculation, and methodology of the survey have been previously described elsewhere [21].

### 2.2. Estimating Total Na and K Intake

Dietary information was obtained from a subsample of participants of ENSANUT-2016: 535 pre-school children (1 to <5 years), 1101 school-age children (5 to <12 years), 1284 adolescents (12 to <20 years), and 1421 adults (>20 years). Those participants with extreme energy intake (outside three standard deviations of the log of energy intake-to-energy requirement ratio) were excluded from the study, as well as pregnant, lactating females, and all individuals ≤1 year old (*n* = 122).

Trained interviewers obtained a 24-h recall from the participants [22]. The recall was collected between Monday and Sunday to capture the intake variability between weekdays and weekends. Participants younger than 15 years old were assisted by the person who cooked and prepared their meals in the household [23]. Energy and nutrient intakes were estimated using the Mexican Food Database (BAM version 18.1.1, by its Spanish acronym) [24].

Participants reported their dietary intake from the previous days as individual foods, custom recipes (individual ingredients that make up the recipe), and standard recipes (sets of default ingredients that make up a recipe when the informer was not able to provide one). For the present analysis, food recipes were disaggregated into their ingredients. The ENSANUT-2016 dataset does not include Na from salt added at the table but does include Na from salt used in preparations.

We classified all foods and beverages that were reported into 36 groups according to their nutritional profile; if they were frequently consumed by the population, they were considered in a single group (for example, tortilla). The groups are shown in Appendix A. We calculated the total Na and K intake and the contribution (percentage of total Na/K intake) of each food group and ratio of Na to K intake (Na-K).

### 2.3. Food Classification Based on NOVA Processing Level

Based on the NOVA food framework, foods and beverages reported in the 24-h recall were classified as: (1) unprocessed or minimally processed foods (those obtained directly from nature or altered in ways that do not introduce any additional substances but may involve removal of inedible parts); (2) processed culinary ingredients (substances derived from foods or nature through methods such as pressing, refining, grinding, milling, and drying and which are used in culinary preparations); (3) processed foods (those manufactured products made by adding sugar, fat, oil, salt, and/or other culinary ingredients to minimally processed foods); and (4) UPF (manufactured formulations made from substances derived from foods or synthesized from other organic sources, preservatives, and additives) [25,26]. Details of the food and beverage subcategory of each NOVA group are described elsewhere [27].

### 2.4. Sociodemographic Characteristics and Socio-Economic Status

Trained personnel applied questionnaires to participants. The questionnaires were previously validated to collect sociodemographic characteristics. Sociodemographic characteristics such as household characteristics, goods, and available services were used to create a socioeconomic status index (SES) using principal component analysis. The SES was classified into three categories using the 33% and 67% percentiles of the index as cutoff points to create the low, medium, and high strata for SES [28].

### 2.5. Anthropometry

Trained personnel measured participants’ weight, height, and waist circumference using international protocols [21]. Weight in kilograms (kg) and height in meters were used to calculate body mass index (BMI, kg/m^2^). The result was categorized according to the WHO criteria [29]. For participants <19 years, we used STATA macro to analyze survey anthropometric data. For adult participants (>19 years), we used the following classification: normal BMI (18.5–24.9 kg/m^2^), overweight (25.0–29.9 kg/m^2^), obesity (≥30.0 kg/m^2^). Abdominal obesity was defined as a waist circumference of ≥80 cm and ≥90 cm for women and men, respectively [30].

### 2.6. Adult Sample

BP was measured using a digital sphygmomanometer Omron HEM-907 XL following the protocol recommended by the American Heart Association [31]. Adults were classified with hypertension when they had a systolic BP ≥ 130 mmHg and/or a diastolic BP ≥ 80 mmHg and/or when they were under pharmacologic treatment for high BP.

Serum and urine biomarkers: The collection protocols for serum biomarker samples and urine samples were reported elsewhere [32]. Impaired fasting glucose was defined according to the American Diabetes Association classification [33]: prediabetes (fasting glucose ≥ 100 and < 126 mg/dL or HbA1c ≥ 5.7 and <6.5%) or diagnosis made during the survey (fasting glucose ≥ 126 mg/dL or HbA1c ≥ 6.5%). Altered serum lipid levels were considered if cholesterol was >200 mg/dL, triglycerides was >150 mg/dL, c-LDL > 100 mg/dL, and c-HDL < 60 mg/dL [34]. Serum creatinine level was calculated through the Chronic Kidney Disease Epidemiology Collaboration (CKD-EPI) equation [35] based on the glomerular filtration rate (GFR, mL/min/1.73 m^2^) and categorized as follows: normal (≥90); mildly reduced (60–89); moderately reduced (30–59); and severely reduced (15–29) [35].

Smoking, previously diagnosed hypertension, cardiovascular disease, and/or acute myocardial infarction/angina pectoris were self-reported within ENSANUT-2016 using the following questions: “have you smoked?” and “do you currently smoke?”, or “has your doctor ever said you had diabetes or high blood sugar?” Each question was asked for each chronic disease. Those who responded “yes” were defined as self-reported current smoker; ex-smoker; or to have a diagnosis of type 2 diabetes, hypertension, and cardiovascular disease (coronary heart disease and cerebro-vascular disease).

### 2.7. Ethical Considerations

All participants signed the informed consent approved by the Institutional Review Board of the MNIPH. The Ethics and Research Commissions of the MNIPH with the Commission number 1401 approved the original protocol. Bioethics registration 17 CEI00120130424 and COFEPRIS registration CEI 17 007 36.

### 2.8. Statistical Analysis

Means, standard deviations, and confidence intervals (95%CI) of total Na and K intakes were estimated considering age group categories. We also estimated Na and K intakes according to sociodemographic characteristics and BMI. The food groups were ranked according to their contribution to Na and K intake by age group. We classified foods by processing level and eating occasions, stratified by age group, and estimated means and percentage contribution of food groups. The quartiles of Na and K intake were categorized by BMI, waist circumference, BP, lipids, glucose, previous medical diagnosis, and smoking.

We obtained the percentage and 95%CI of the population with a high Na intake, a low K intake, as well as an Na-K ratio > 1.0 according to sociodemographic characteristics. A high level of Na was considered when Na intake was above 800 mg for children 1–3 years; 1000 mg for children 4–8 years; 1200 mg for children 9–13 years, 1500 mg for adolescents 14–18 years; and 2000 mg for adults 19 years and older. K deficiency was considered when the intake of K was less than 2000 mg for children 1–3 years; 2300 mg for children 4–8 years; 2500 mg for male children 9–13 years; 2300 mg for adolescent women ages 9–18; 3000 mg for male adolescent ages 14–18; and 3510 mg for adults ages 19 and older.

To explore differences across categories, we used linear regression models and design-based Wald statistics for quantitative variables. We employed the design-based F-statistic Pearson Chi-square and logistic regression for categorical data. A general *p* < 0.05 value was considered to set the statistical significance. All the analyses were performed using the SVY command to consider the survey sample weights and the sample design of the ENSANUT-2016. All the analyses were carried out in STATA version 15 (College Station, TX, USA). Stata Corp. Release 14, vol. 1–4. College Station (TX): Stata Press.

## 3. Results

Data from 4219 participants were analyzed, representing more than 136 million Mexicans. Table 1 shows Na and K intake according to age group and sociodemographic characteristics. The mean intake of Na in pre-school children was 1512 mg (95%CI: 1377, 1647); in school-age children it was 2844 mg (95%CI: 2379, 3309); in adolescents (the age group with the highest intake), 3743 mg (95%CI: 3144, 4341); and in adults it was 3132 mg (95%CI: 2794, 3470). The mean intake of K in pre-schoolchildren was 1616 mg (95%CI: 1481, 1751); in school-age children, 2256 (95%CI: 1987, 2525); in adolescents, 2967 mg (95%CI: 2612, 3322); and in adults, 3401 mg (95%CI: 2862, 3939). The Na-K ratio was higher in school-age children (1.4, 95%CI: 1.1, 1.7) and adolescents (1.3, 95%CI: 1.3, 1.4).

### 3.1. Food Groups That Contribute to Sodium and Potassium Intake

Table 2 shows the 10 food groups that contribute the most Na and K to Mexicans’ dietary intake, categorized by age groups. The food groups that had the highest contribution to daily Na intake in preschool children were: dairy (197 mg), processed meats (148 mg), and condiments (77 mg); in school-age children: dairy (229 mg), processed meats (237 mg), and salty snacks (117 mg); in adolescents: dairy (287 mg), processed meats (269 mg), and salty snacks (172 mg); and in adults: red meat (197 mg), dairy (184 mg), and processed meat (179 mg). Also, by age group, the food categories that contributed the most to K intake in pre-school children were: dairy (397 mg), fruits (261 mg), and vegetables (169 mg); in school-age children they were: vegetables (350 mg), dairy (324 mg), fruits (279 mg), and corn tortillas (216 mg); in adolescents they were: fruits (415 mg), vegetables (386 mg), and corn tortillas (363 mg); and in adults they were: vegetables (702 mg), fruits (467 mg), and corn tortillas (401 mg).

### 3.2. Contribution of Na and K According to the NOVA Classification

Figure 1 shows the contribution of Na intake according to the NOVA classification in the Mexican population. Processed and UPF contributed 49% of Na intake in preschool children and school-age children; 50% for adolescents; and 39% for adults. Figure 2 shows the contribution of K intake according to the NOVA classification in the Mexican population. Potassium intake was lower for processed and UPF, contributing 31% of K intake for preschool children; 23% for school-age children; 22% for adolescents; and 14% for adults (Figure 2). The mean intake and contribution of Na and K according to the NOVA classification in the Mexican population is displayed in Appendix A.

### 3.3. Na and K Intake: Health Risk

Table 3 shows the proportion of participants with high Na and K intake, as well as a Na-K ratio greater than 1.0. Among preschool children, 65% (95%CI: 51.7–76.9) had high SES, while 98% (95%CI: 90.3–99.6) of those who were overweight showed insufficient K. In addition, the lowest proportion with Na-K ratio > 1.0 was present in the southern region of Mexico (35.2%, 95%CI: 26.1–45.6). Among school-age children, a higher proportion of children with high Na intake was observed in urban areas (84.8%, 95%CI: 79.5–89.0), high SES (87.2%, 95%CI: 79.9–92.1), as well as among those children with obesity (88.0%, 95%CI: 80.0–93.1). The lowest proportion with insufficient K and a Na-K ratio > 1.0 was present in schoolchildren with obesity (60.3%, 95%CI: 48.7–70.8) and in those located in the southern region of Mexico (49.3%, 95%CI: 40.8–57.7), respectively. Among adolescents, a higher proportion of high Na intake was observed in those participants with overweight (82.4%, 95%CI: 75.8–87.5) and obesity (86.8%, 95%CI: 72.7–94.2). The lowest proportion with insufficient K was found in those located in the central region of Mexico (47.9%, 95%CI: 36.4–59.5). Among adults, a higher proportion of people with high SES consumed high amounts of Na (70.6%, 95%CI: 60.7–78.9).

### 3.4. Na and K Intake: Nutrition Status and Clinical Characteristics

Table 4 shows the quartiles of Na and K intake in adults according to the participants’ nutrition status and clinical characteristics. Adults in the fourth quartile of sodium intake (5049.6 mg/day) had lower cholesterol serum concentrations (181.4 mg/dL) and HDL-c (35.5 mg/dL) than adults in the first quartile (cholesterol 202.5 mg/dL and HDL-c 40.8 mg/dL) (*p* < 0.005). Adults who were in the fourth quartile showed lower levels of HDL-c (34.3 mg/dL) compared to those in the first quartile (41.6 mg/dL) (*p* < 0.005). Participants in the third quartile of K intake (2166.5 mg/day) had higher serum concentrations of LDL-c (123.5 mg/dL) compared to those in the first quartile (104.9 mg/dL) (*p* < 0.005).

## 4. Discussion

This study shows that the Na dietary intake in Mexicans exceeds the WHO recommendation (2 g/day) for preschool children, school-age children, adolescents, and adults. The main food sources of dietary Na were salt, cereals, dairy, and processed and red meats. For dietary K, all age groups consumed insufficient amounts of K according to the WHO and IOM recommendations. The main food sources of K were vegetables, dairy, fruits, tortillas, and legumes. In addition, processed and UPF and beverages contributed the most to dietary Na, while minimally processed or unprocessed foods contributed the most K in the diet. In addition, adults who consumed higher amounts of Na and K have a lower serum concentration of cholesterol, HDL-c, and LDL-c levels.

The Na intake found in this study was similar to that found in Latin American and the Caribbean countries, where the average Na intake is close to 3.4 g/day [36], observing higher intake in Brazil, Chile, and Colombia (>4.7 g/day). It is even similar to worldwide Na intake (about 3.9 g/day) [37]. Potassium intake is low in this population according to recommendations from international organizations [6,7]. Globally, it has been shown that the intake of this nutrient has increased in recent years, slightly exceeding the expectations (3.7 g/day) [38]. However, in countries such as Brazil, slightly low potassium intake has been identified, similar to the adult population in this study [39].

The intake of both nutrients may be due, in a large extent, to modified nutritional behavior [40,41,42]. Most countries have experienced a change in their traditional diets, shifting towards a western diet, which is characterized by increased intake of processed foods [43] and decreased intake of fruits and vegetables [44]. If this nutrient intake is maintained, it will continue to contribute to the main cause of mortality in Mexico and the years of life lost, also worsening years of life lost due to disability [45].

Our study showed that processed and UPF and beverages contribute the most to dietary Na, while minimally processed or unprocessed contribute the most K. This is expected since the higher the degree of processing, the greater the amount of critical nutrients that are added (such as sodium), while the lower the processing, the greater the amount of fiber due to the natural state of the product [25].

A study carried out in Canada found that the population that consumed a proportion of processed foods in their diet had a higher Na content and a lower K content; the opposite was observed in a diet that consisted of either minimally processed or unprocessed foods [46]. Similarly, a study in Australia found that individuals with higher intake of processed foods had more Na and less K in their diet, compared to those who have a low intake, which contributes more K and less Na in their diet [47]. Furthermore, this study showed that foods that are minimally processed contribute more than 50% of dietary K. If this trend continues, it might decrease K intake in our population and increase the Na-K ratio, posing a higher cardiovascular risk [48]. However, the studies that link NOVA classified foods with health events or diet quality do not use added salt, since this would contribute greatly to the group of culinary ingredients and would be the first source of sodium in the diet [49,50]. In this study, we include this source of Na (salt in preparations) within the culinary ingredients in the NOVA classification; however, processed and UPF provide between 40–50% of Na among the different age groups.

Our results showed an Na intake of 1512 mg/day and a K intake of 1616 mg/day for pre-schoolchildren. In Mexicans, the amount of Na is high for the age, however, the Japanese population (2300 mg) [51], Polish (1220 mg) [52], and Australian (3400 mg) [53] consume more Na. For K intake, it is higher in Japan (1700 mg) [51] and lower in Poland (947 mg) [52] and Australia (1119 mg) [53]. This may be due to eating patterns between countries. Also, the intake of processed foods may be greater in developing countries. Mexico exceeds amounts recommended by the WHO and the IOM, probably due to poor diet quality [54].

In school-age children, Na intake exceeded the WHO recommendation, while K intake was below the recommended level; the Na-K ratio was 1.4 (95%CI: 1.1, 1.7). These results coincide with those obtained in Indonesian children between 9 and 12 years, who had a high Na intake (>2300 mg/day) and a low K intake (<2500 mg/day) [55]. On the other hand, in Spanish children between 9 and 13 years old, Na intake was close to 2500 mg/day, while K intake was 2800 mg/day [56]. Guatemalan children (6–11 years) reported a low intake of Na (831 mg/day), as well as a low intake of K (1364 mg/day) [57]. These similarities may occur because in these countries the consumption of fruits and vegetables containing K is low, while the consumption of processed foods and UPF that contain Na is high. The main food groups that contribute to Na intake in this population group are salt, cereals, dairy products, processed meats, salty snacks, red meats, and seasonings. Food groups such as dairy, vegetables, fruits, as well as corn tortillas and root vegetables contribute the most K to the diet. These results are consistent with those obtained by Grimes et al., since they showed that in Australian children between 4 and 12 years, the main food groups that contributed to a high Na intake in the diet were cereals, as well as meat, poultry, condiments or seasonings, and flavored sauces. Regarding K intake, this population showed that milk products and dishes, as well as vegetables, fruits, meats, and cereals contribute most of the intake of this nutrient [58]. On the other hand, Cuadrado et al., showed that the main food sources that contributed Na to the diet in Spanish children between 7 and 11 years old were meats and meat products, cereals, grains and legumes, as well as pre-cooked and ready-to-eat meals [49]. This may also be due to the fact that non-basic foods that are high in fat and sugar, animal products, and milk and dairy derivatives were found to contribute 22% to total dietary energy in schoolchildren [43].

In adolescents, the mean dietary Na intake was higher, and the mean dietary K intake was below the levels recommended by international agencies. These results were consistent with international studies; for example, Portuguese adolescents showed a high mean dietary Na intake (3500 mg/day), while the mean dietary K intake was low (2150 mg/day) [59]. Furthermore, adolescents in Morocco reported consuming more than 2134 mg/day [57], while adolescents in China reported consuming a median intake of dietary Na of 4300 mg/day while consuming 1600 mg/day of dietary K [60]. In this population group, the main sources of foods that contributed Na were salt, cereals, dairy products, processed meats, salty snacks, and red meat. The main food sources that contributed a high K intake were fruits, vegetables, corn tortillas, dairy, legumes, and red meat [61]. These results were consistent with that of American adolescents since the main food sources that contributed to high Na intake were mixed dishes—pizza, mixed-Mexican dishes, as well as mixed avocado sandwiches, breads, rolls, and tortillas [61]. On the other hand, the main food sources that contributed to K intake were milk, white potatoes, fruits, 100% juices, poultry, and mixed-Mexican dishes [62]. This may be because this population group is the one that consumes the least fruits and vegetables compared to adults and those under 12 years of age. In addition, adolescents have been identified to have a Western diet pattern, which is characterized by high processed food and salt intake [43].

Mexican adults have a high Na intake (3132 mg/day), as well as a low K intake (3400 mg/day). The previous results were consistent with previous studies obtained from the SALMEX cohort in Mexico City [9]. This study showed that the estimated Na intake was high (2600 mg/day), while the (urinary) K intake was low (1982 mg/day) [9]. A study carried out in China showed that Na intake in the adult population was above 4100 mg/day, while K intake was between 1500 and 1600 mg/day [63]. Another study carried out with American adults showed that the mean population of Na intake was close to 3500 mg/day, while the mean K intake was close to 2800 mg/day [64]. Our results did not show a high proportion between Na-K ratio; however, other countries have shown that these proportions are above two [63,65], which was related to diastolic and systolic pressure. Mexican adults do not have enough K intake to comply with the WHO recommendations. K intake is higher than in other countries like China [63], probably because intake of non-basic foods in the adult diet is lower than that of minors [43].

Our results showed that the food groups that most contribute to Na intake are salt, cereals, red meat, dairy, processed meats, as well as seasonings, while the food groups that contribute the most to K intake are vegetables, fruits, corn tortillas, legumes, dairy, as well as root vegetables. These results are not very consistent with those obtained in a subsample of the SALMEX cohort, a local study that only includes a region of the country [20]. This may be due to the fact that our results have national coverage and there is a greater variety of food intake in different regions. A study conducted with Chinese adults showed that the main source of Na distribution was salt (69%), as well as soy sauce (8%) and processed food (6%) [66]. However, the Mexican diet is different from the Chinese; also, China has higher mortality rates attributed to high Na intake compared to Mexico [45]. In addition to the above, our results are consistent with a study conducted with Australian adults, in which the main sources of K intake were vegetables, meat, poultry, fruits, and milk products [65].

This study showed a relationship between higher sodium intake (fourth quartile) and cholesterol and LDL-c levels, compared to people who consumed less sodium (first quartile). These results are consistent with a review of randomized studies by Graudal et al. who showed an inverse relationship between sodium intake and serum cholesterol concentrations compared to those with high Na intake [67]; in addition, this review shows that decreasing sodium intake increases serum concentrations of undesirable lipids (Cholesterol, HDL-c, and LDL-c) [67]. In Denmark, it was found that salt intake is negatively associated with HDL-c but positively associated with triglycerides [68]. These associations should be viewed and interpreted with caution, since multiple biases influence the measure that comes from indirect methods that estimate sodium intake [69].

Finally, it is important to highlight that about 49% of Mexican adults have hypertension [70] and that the main cause of death is cardiovascular disease [71]. Our studies indicate that a large proportion of adults have a high intake of Na (64%) and an insufficient intake of K (65%). These findings indicate that the risk of suffering from hypertension or suffering from some complication is greater. Simulation studies carried out in Latin America have estimated that reducing sodium consumption as recommended by the WHO could reduce about 47,000 deaths from cardiovascular diseases, mainly from coronary, hypertensive, and cardiovascular disease, the equivalent of 85 million dollars in health care [72,73].

The main strengths of this study are that it is based on a representative sample, it includes all age groups nationwide, and is the first to identify Na and K intake as well as the main dietary sources of Na and K (among them the nova classification). The detailed dietary intake data were collected at the brand level for each food consumed, and Na values were updated with analytical data on Mexican foods. In addition, we utilized nutrition status and clinical characteristics biomarkers with a high precision in adults.

The main limitation of this study is that we estimate intake with a 24-h food recall, and this information does not come from a direct biomarker such as 24-h urine (reference method). The 24-h food recall tends to underestimate the intake of nutrients such as Na and K due to the lack of precision and memory of the participant, since this method is complex, requires a lot of work for both participants and field workers, and participants tend to change the consumption report in the dietary recall interview [69]. Furthermore, this study did not consider Na added at the table, but did measure Na used in food preparation, allowing for an even greater underestimation.

Furthermore, Na intake has been found to be highly correlated with total energy intake, due to the wide variety of foods and meals [69]. Likewise, it has been identified that Na consumption is higher in those with high BMI. In Mexico, 30% of children under 12 years of age, 36% of adolescents [74], and 70% of adults [75] are overweight or obese. Therefore, Na consumption is likely to be underreported.

Despite the above, the population information generated in this research is essential to have an approach in the consumption of Na and K in Mexico, which could be used for the design and evaluation of strategies to reduce the intake of Na and increase K intake in this country.

## 5. Conclusions

Our results show that all age groups have high Na and low K intakes, as well as an inadequate Na-K ratio according to international recommendations.

Also, there is a relationship between higher sodium intake and low cholesterol, HDL-c, and LDL-c levels, compared to people who consume less sodium.

In Mexico, it is important to promote a diet from an early age with foods rich in K, such as: fruits, vegetables, legumes, and milk. Although these foods are the main sources of K intake in this population; their current intake seems insufficient to provide enough K in the diet. The results of this article highlight the importance of implementing reformulation strategies in processed and UPF and beverages because they contribute the most to dietary Na. If this trend continues, the probability of having an unhealthy Na-K ratio will increase, along with cardiovascular risk.

## Figures and Tables

**Figure 1 nutrients-14-00281-f001:**
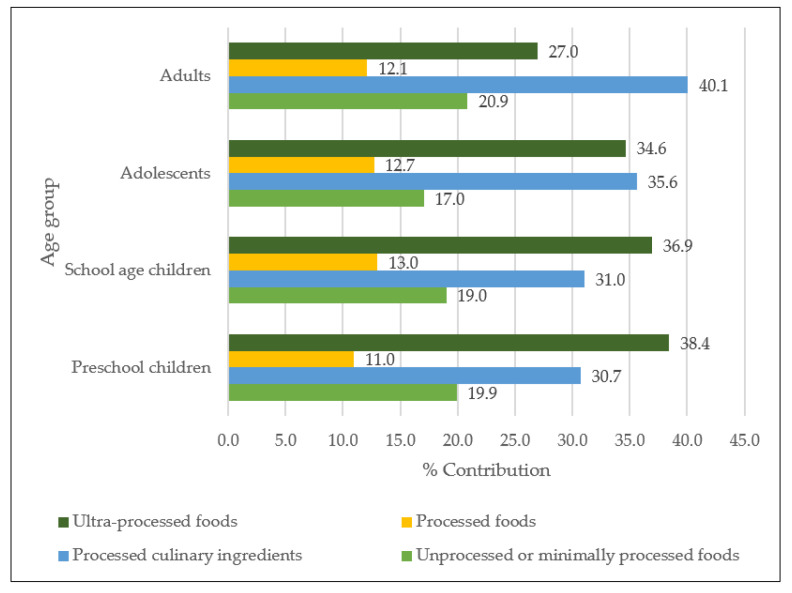
Percentage contribution of sodium intake by NOVA classification in the Mexican population: ENSANUT-2016.

**Figure 2 nutrients-14-00281-f002:**
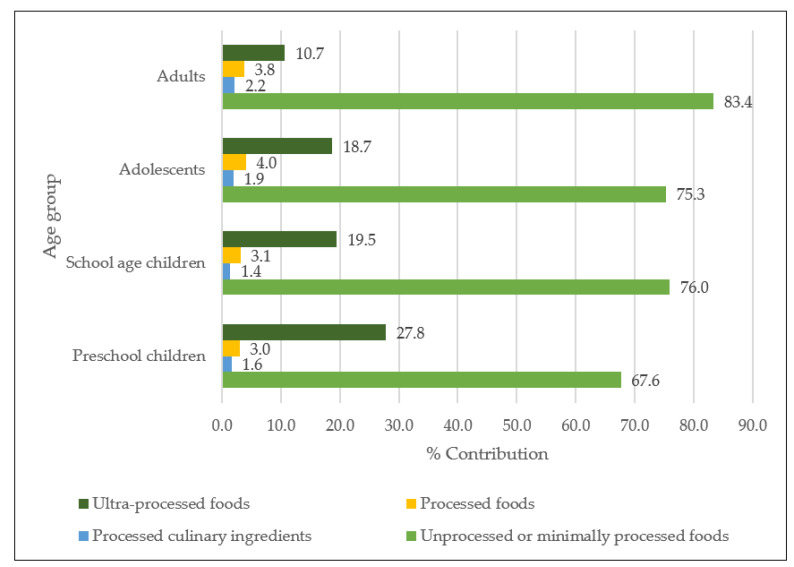
Percentage contribution of potassium intake by Nova classification in the Mexican population: ENSANUT-2016.

**Table 1 nutrients-14-00281-t001:** Sodium and potassium mean intake by age group according to sociodemographic characteristics and Body Mass Index in Mexican population. ENSANUT 2016.

	Preschool Children		School-Age Children		Adolescents		Adults	
	*n* = 528	N = 8,584,831	*n* = 1095	N = 16,144,480	*n* = 1240	N = 23,988,992	*n* = 1356	N = 87,921,191
	Mean (SE)	95%CI	Mean (SE)	95%CI	Mean (SE)	95%CI	Mean (SE)	95%CI
**Total** (*n* = 4219; N = 136,639,494)								
Sodium intake (mg/day)	1512.2 ± 68.6	(1377.1, 1647.2)	**2843.8 ± 236.1**	**(2379.0, 3308.6)**	**3743.2 ± 304.2**	**(3144.4, 4341.9)**	**3132.3 ± 171.9**	**(2794.0, 3470.7)**
Potassium intake (mg/day)	1615.7 ± 68.6	(1480.6, 1750.7)	**2255.7 ± 136.5**	**(1986.9, 2524.5)**	**2966.6 ± 180.3**	**(2611.7, 3321.5)**	**3400.6 ± 273.5**	**(2862.1, 3939.0)**
Na-K ratio	1.1 ± 0.1	(0.9, 1.2)	**1.4 ± 0.1**	**(1.1, 1.7)**	**1.3 ± 0.0**	**(1.3, 1.4)**	1.1 ± 0.0	(1.0, 1.2)
**Sex**								
Women (*n* = 2312; N = 72,673,426)								
Sodium intake (mg/day)	1494.8 ± 95.6	(1306.7, 1682.9)	2817.2 ± 425.6	(1979.5, 3655.0)	3286.6 ± 240.2	(2813.8, 3759.5)	2927.9 ± 217.2	(2500.3, 3355.5)
Potassium intake (mg/day)	1594.4 ± 93.0	(1411.3, 1777.6)	2052.2 ± 77.6	(1899.4, 2205.1)	2560.6 ± 138.5	(2288.0, 2833.1)	3250.5 ± 427.4	(2409.1, 4091.9)
Na-K ratio	1.0 ± 0.1	(0.9, 1.1)	1.6 ± 0.3	(1.0, 2.2)	1.4 ± 0.1	(1.2, 1.5)	1.1 ± 0.1	(1.0, 1.2)
Men (*n* = 1907; N = 63,966,069)								
Sodium intake (mg/day)	1528.6 ± 109.7	(1312.5, 1744.6)	2867.2 ± 253.4	(2368.3, 3366.1)	4236.6 ± 567.5	(3119.5, 5353.7)	3383.7 ± 257.2	(2877.3, 3890.1)
Potassium intake (mg/day)	1635.8 ± 96.6	(1445.6, 1826.0)	2434.6 ± 236.4	(1969.2, 2900.0)	**3405.5 ± 329.3**	**(2757.1, 4053.9)**	3585.0 ± 325.7	(2943.7, 4226.3)
Na-K ratio	1.1 ± 0.1	(0.9, 1.4)	1.3 ± 0.1	(1.2, 1.4)	1.3 ± 0.1	(1.2, 1.5)	1.1 ± 0.1	(1.0, 1.2)
**Area of residence**								
Rural (*n* = 2227; N = 37,106,754)								
Sodium intake (mg/day)	1439.1 ± 107.6	(1225.8, 1652.3)	2515.5 ± 243.5	(2032.9, 2998.1)	3498.7 ± 202.9	(3096.6, 3900.8)	3133.7 ± 375.8	(2389.0, 3878.5)
Potassium intake (mg/day)	1552.4 ± 100.4	(1353.5, 1751.3)	2193.2 ± 125.1	(1945.2, 2441.1)	2817.7 ± 135.3	(2549.5, 3085.9)	3626.9 ± 452.4	(2730.3, 4523.5)
Na/-K ratio	1.0 ± 0.1	(0.9, 1.1)	1.3 ± 0.1	(1.1, 1.5)	1.3 ± 0.1	(1.2, 1.4)	1.0 ± 0.1	(0.9, 1.2)
Urban (*n* = 1992; N = 99,532,740)								
Sodium intake (mg/day)	1538.7 ± 84.6	(1371.7, 1705.7)	2978.9 ± 317.6	(2352.0, 3605.9)	3824.4 ± 398.5	(3037.9, 4610.9)	3131.8 ± 189.5	(2757.7, 3505.9)
Potassium intake (mg/day)	1638.7 ± 85.9	(1469.0, 1808.3)	2281.4 ± 185.6	(1915.1, 2647.7)	3016.1 ± 233.3	(2555.6, 3476.6)	3314.9 ± 337.7	(2648.4, 3981.5)
Na-K ratio	1.1 ± 0.1	(0.9, 1.3)	1.5 ± 0.2	(1.1, 1.9)	1.4 ± 0.1	(1.2, 1.5)	1.1 ± 0.1	(1.0, 1.2)
**Socioeconomic tertile**								
Low (*n* = 1464; N = 31,441,350)								
Sodium intake (mg/day)	1388.4 ± 112.4	(1166.7, 1610.2)	2368.8 ± 201.1	(1972.1, 2765.5)	2896.1 ± 144.7	(2610.7, 3181.6)	2854.5 ± 278.6	(2304.9, 3404.2)
Potassium intake (mg/day)	1483.7 ± 101.0	(1284.4, 1683.0)	2163.3 ± 250.4	(1669.3, 2657.3)	2701.7 ± 134.7	(2436.1, 2967.4)	3283.7 ± 261.1	(2768.6, 3798.8)
Na-K ratio	1.0 ± 0.1	(0.9, 1.2)	1.3 ± 0.1	(1.1, 1.4)	1.2 ± 0.0	(1.1, 1.3)	1.0 ± 0.1	(0.8, 1.1)
Medium (*n* = 1516; N = 40,785,812)								
Sodium intake (mg/day)	1448.3 ± 108.6	(1234.3, 1662.2)	2958.1 ± 574.0	(1827.6, 4088.7)	4324.4 ± 772.5	(2802.8, 5846.0)	3264.8 ± 379.6	(2517.0, 4012.6)
Potassium intake (mg/day)	1637.2 ± 118.2	(1404.3, 1870.1)	1982.8 ± 116.5	(1753.3, 2212.4)	3423.6 ± 445.3	(2546.4, 4300.8)	3450.6 ± 486.4	(2492.6, 4408.7)
Na-K ratio	1.1 ± 0.2	(0.8, 1.4)	1.7 ± 0.4	(0.9, 2.5)	1.3 ± 0.1	(1.1, 1.5)	1.2 ± 0.1	(1.0, 1.3)
High (*n* = 1239; N = 64,412,333)								
Sodium intake (mg/day)	1684.5 ± 126.3	(1435.7, 1933.3)	3053.3 ± 313.5	(2435.7, 3670.9)	**3707.6 ± 271.0**	**(3173.6, 4241.5)**	3186.0 ± 216.4	(2759.6, 3612.4)
Potassium intake (mg/day)	1692.7 ± 123.8	(1448.7, 1936.6)	2546.4 ± 264.9	(2024.6, 3068.3)	2745.9 ± 132.0	(2485.8, 3005.9)	3426.1 ± 463.0	(2513.8, 4338.4)
Na-K ratio	1.1 ± 0.1	(1.0, 1.3)	1.3 ± 0.1	(1.1, 1.4)	**1.5 ± 0.1**	**(1.3, 1.6)**	1.1 ± 0.1	(1.0, 1.3)
**Body Mass Index ^a^**								
Normal (*n* = 2277; N = 49,562,419)								
Sodium intake (mg/day)	1477.9 ± 70.7	(1338.6, 1617.1)	2533.5 ± 327.7	(1888.3, 3178.7)	2894.7 ± 121.6	(2655.2, 3134.1)	2898.6 ± 234.6	(2436.6, 3360.5)
Potassium intake (mg/day)	1576.3 ± 73.3	(1431.9, 1720.6)	1878.3 ± 77.0	(1726.8, 2029.9)	2380.0 ± 93.7	(2195.5, 2564.5)	3072.9 ± 285.5	(2510.8, 3635.0)
Na-K ratio	1.1 ± 0.1	(0.9, 1.2)	1.5 ± 0.2	(1.1, 1.9)	1.4 ± 0.1	(1.2, 1.5)	1.1 ± 0.1	(0.9, 1.3)
Overweight (*n* = 949; N = 35,937,427)								
Sodium intake (mg/day)	**1097.8 ± 128.3**	**(845.2, 1350.4)**	2376.3 ± 139.2	(2102.2, 2650.4)	2856.8 ± 138.7	(2583.6, 3129.9)	3110.9 ± 318.0	(2484.5, 3737.3)
Potassium intake (mg/day)	**1064.5 ± 153.1**	**(762.9, 1366.1)**	2103.3 ± 106.4	(1893.7, 2312.9)	2188.1 ± 108.7	(1973.9, 2402.2)	3163.4 ± 400.8	(2374.1, 3952.7)
Na-K ratio	1.1 ± 0.1	(0.8, 1.3)	1.2 ± 0.1	(1.1, 1.4)	1.4 ± 0.1	(1.3, 1.5)	1.1 ± 0.1	(1.0, 1.3)
Obesity (*n* = 840 N = 42,639,888)								
Sodium intake (mg/day)	**2194.8 ± 254.5**	**(1693.5, 2696.2)**	2720.2 ± 224.5	(2277.9, 3162.5)	3359.6 ± 375.0	(2620.7, 4098.4)	3287.1 ± 257.1	(2780.5, 3793.6)
Potassium intake (mg/day)	**2102.1 ± 131.4**	**(1843.1, 2361.0)**	2095.6 ± 168.8	(1763.1, 2428.1)	2653.8 ± 252.6	(2156.2, 3151.5)	3786.6 ± 511.3	(2779.3, 4793.9)
Na-K ratio	1.1 ± 0.2	(0.8, 1.5)	1.5 ± 0.1	(1.2, 1.7)	1.3 ± 0.2	(1.0, 1.6)	1.0 ± 0.1	(0.9, 1.2)

*n*, sample size; N, expanded sample. **Bold** numbers mean statistically significant difference vs reference category (*p* < 0.05). The reference category is the first row of each variable, except for the first variable where it is the first column. ^a^ Body mass index (BMI): <25 kg/m^2^ (normal); 25–29.9 kg/m^2^ (overweight); ≥30 kg/m^2^ (obesity). For those under 19 years of age, the BMI for age was used according to the WHO child growth patterns.

**Table 2 nutrients-14-00281-t002:** Dietary top 10 food and beverage groups contributing to sodium and potassium intake in Mexican population: ENSANUT 2016.

	Preschool Children			School-Age Children			Adolescents			Adults		
**Sodium**												
**Ranking**	**Food Groups**	**mg/day**	**% Contribution**	**Food Groups**	**mg/day**	**% Contribution**	**Food Groups**	**mg/day**	**% Contribution**	**Food Groups**	**mg/day**	**% Contribution**
**1**	Salt	399.3 ± 41.3	26.8 ± 1.5	Salt	1016.9 ± 216.2	28.1 ± 1.2	Salt	1186.2 ± 133.7	32.2 ± 2.3	Salt	1153.5 ± 104.7	36.5 ± 1.5
**2**	Cereals	253.6 ± 38.7	13.4 ± 1.6	Cereals	458.1 ± 43.3	16.0 ± 1.2	Cereals	641.7 ± 117.0	13.6 ± 1.4	Cereals	448.5 ± 60.5	12.5 ± 1.1
**3**	Dairy	196.8 ± 12.3	16.1 ± 1.3	Dairy	228.8 ± 20.0	10.2 ± 0.6	Dairy	287.1 ± 44.3	8.4 ± 1.0	Red meat	197.3 ± 40.6	5.9 ± 0.9
**4**	Processed meats	148.1 ± 28.1	7.6 ± 1.2	Processed meats	237.1 ± 34.7	7.9 ± 0.8	Processed meats	268.7 ± 43.5	6.6 ± 0.9	Dairy	184.0 ± 16.9	6.7 ± 0.5
**5**	Seasonings	76.6 ± 20.1	4.4 ± 1.0	Salty snacks	116.9 ± 19.3	4.1 ± 0.4	Salty snacks	172.3 ± 34.4	4.0 ± 0.6	Processed meats	178.6 ± 31.6	4.5 ± 0.6
**6**	Eggs	42.7 ± 3.7	3.3 ± 0.3	Red meat	105.4 ± 17.0	3.6 ± 0.5	Red meat	159.0 ± 31.8	4.7 ± 1.2	Seasonings	125.0 ± 44.2	3.5 ± 1.0
**7**	Salty snacks	45.5 ± 15.4	2.3 ± 0.6	Seasonings	104.1 ± 30.2	2.7 ± 0.5	Seasonings	89.02 ± 19.1	2.5 ± 0.4	Corn tortilla	95.5 ± 8.6	4.4 ± 0.4
**8**	Red meat	27.8 ± 8.4	1.7 ± 0.4	Corn tortilla	52.2 ± 4.8	2.9 ± 0.2	Corn tortilla	87.6 ± 9.4	3.5 ± 0.3	Eggs	68.0 ± 8.9	2.9 ± 0.4
**9**	Cereal-based sweets	26.1 ± 6.1	1.8 ± 0.4	Eggs	51.6 ± 4.2	2.9 ± 0.3	Sweet bakery bread	80.3 ± 14.6	3.2 ± 0.6	Sweet bakery bread	63.4 ± 8.4	2.9 ± 0.4
**10**	Non-cereal-based sweets	26.5 ± 4.6	1.8 ± 0.2	R-to-E Cereals	51.1 ± 11.0	2.7 ± 0.4	Eggs	77.7 ± 9.1	2.7 ± 0.3	Cereals based sweets	60.3 ± 18.1	1.9 ± 0.6
**Potassium**												
**Ranking**	**Food groups**	**mg/day**	**% Contribution**	**Food groups**	**mg/day**	**% Contribution**	**Food groups**	**mg/day**	**% Contribution**	**Food groups**	**mg/day**	**% Contribution**
**1**	Dairy	396.7 ± 32.3	23.7 ± 1.7	Vegetables	350.3 ± 42.5	13.5 ± 0.9	Fruits	414.5 ± 67.7	11.7 ± 1.2	Vegetables	701.6 ± 124.8	17.7 ± 1.2
**2**	Fruits	260.7 ± 28.5	14.7 ± 1.2	Dairy	324.2 ± 29.1	15.6 ± 0.9	Vegetables	385.7 ± 46.7	13.2 ± 1.1	Fruits	467.0 ± 74.2	12.3 ± 0.8
**3**	Vegetables	168.7 ± 19.1	9.8 ± 0.8	Fruits	278.5 ± 22.3	12.1 ± 0.8	Corn tortilla	363.3 ± 38.9	13.8 ± 0.9	Corn tortilla	401.4 ± 35.7	13.9 ± 1.0
**4**	Yogurt and milk-based drinks	101.4 ± 14.5	5.9 ± 0.8	Corn tortilla	215.9 ± 19.9	10.8 ± 0.6	Dairy	360.4 ± 48.4	11.5 ± 0.9	Legumes	302.6 ± 38.1	9.3 ± 0.9
**5**	Corn tortilla	89.3 ± 9.1	6.4 ± 0.6	Legumes	149.8 ± 18.0	6.7 ± 0.5	Legumes	273.6 40.6	8.2 ± 0.8	Dairy	233.2 ± 26.6	7.9 ± 0.6
**6**	Legumes	81.4 ± 10.9	5.3 ± 0.6	Root vegetables	134.2 ± 34.1	4.2 ± 0.6	Red meat	161.4 ± 20.8	5.7 ± 0.4	Root vegetables	198.6 ± 47.8	4.5 ± 0.6
**7**	SSBs carbonated	75.6 ± 11.5	4.8 ± 0.8	Red meat	101.6 ± 11.7	4.8 ± 0.6	Cereals	100.2 ± 11.3	4.0 ± 0.4	Red meat	172.0 ± 24.3	5.9 ± 0.5
**8**	Root vegetables	59.4 ± 13.7	3.0 ± 0.5	Cereals	88.2 ± 6.7	4.5 ± 0.2	Salty snacks	98.2 ± 19.0	3.1 ± 0.5	Cereals	138.9 ± 27.7	4.2 ± 0.4
**9**	Cereals	55.8 ± 7.3	4.3 ± 0.5	Yogurt and milk-based drinks	82.1 ± 14.9	3.0 ± 0.4	Poultry	95.4 ± 23.7	3.4 ± 0.7	Coffee and tea	134.0 ± 36.9	3.5 ± 0.4
**10**	Poultry	45.2 ± 6.2	3.0 ± 0.4	Poultry	79.7 ± 11.4	3.3 ± 0.3	Root vegetables	88.7 ± 16.01	2.9 ± 0.4	Poultry	120.8 ± 21.1	4.2 ± 0.5

Salt refers to that reported in the preparations. SSBs, sugar-sweetened beverages. R-to-E Cereals, ready-to-eat cereals.

**Table 3 nutrients-14-00281-t003:** Proportions of high sodium, insufficient potassium intake, and sodium–potassium ratio by age group in Mexican population. ENSANUT-2016 ^1,2,3^.

	Preschool Children			School-Age Children			Adolescents			Adults		
	High Na	Insufficient K	Na-K	High Na	Insufficient K	Na-K	High Na	Insufficient K	Na-K	High Na	Insufficient K	Na-K
**Total** (*n* = 4219; N = 136,639,494)												
**Age**	73.6 (66.9, 79.4)	73.3 (66.2, 79.4)	41.5 (34.6, 48.8)	**82.1 (78.1, 85.6)**	67.1 (61.5, 72.3)	**61.1 (56.3, 65.8)**	81.6 (76.3, 85.9)	**58.0 (49.1, 66.3)**	**62.5 (52.6, 71.5)**	**64.1 (57.1, 70.5)**	65.6 (56.7, 73.5)	45.7 (38.6, 53.0)
**Sex**												
Women	74.8 (67.0, 81.2)	71.9 (61.6, 80.2)	44.8 (35.3, 54.7)	80.9 (75.4, 85.4)	66.8 (59.8, 73.2)	59.5 (53.3, 65.5)	79.4 (70.5, 86.2)	62.1 (49.5, 73.3)	60.5 (44.2, 74.7)	60.9 (51.0, 70.0)	69.7 (55.0, 81.3)	45.1 (35.0, 55.6)
Men	72.5 (61.9, 81.1)	74.7 (64.5, 82.8)	38.4 (28.6, 49.2)	83.2 (77.0, 88.0)	67.4 (59.0, 74.9)	62.6 (54.8, 69.7)	84.0 (76.8, 89.3)	53.5 (42.0, 64.7)	64.8 (52.6, 75.3)	68.0 (58.6, 76.1)	60.4 (49.5, 70.4)	46.4 (37.3, 55.8)
**Area of residence**												
Rural	71.8 (63.3, 79.0)	80.5 (72.8, 86.5)	40.3 (31.4, 49.9)	75.6 (69.9, 80.5)	67.1 (59.7, 73.8)	52.5 (46.0, 58.9)	79.3 (73.5, 84.1)	54.0 (45.6, 62.2)	60.2 (53.7, 66.4)	65.6 (56.1, 74.0)	61.5 (47.9, 73.6)	37.8 (27.7, 49.1)
Urban	74.3 (65.49,81.5)	70.7 (61.4, 78.6)	41.9 (33.1, 51.3)	**84.8 (79.5, 89.0)**	67.1 (59.7, 73.8)	**64.7 (58.5, 70.4)**	82.4 (75.5, 87.7)	59.3 (47.8, 69.8)	63.3 (50.0, 74.8)	63.5 (54.5, 71.6)	67.1 (55.8, 76.7)	48.7 (39.6, 57.8)
**Region**												
North	73.2 (58.7, 84.1)	70.8 (50.9, 85.1)	55.4 (38.6, 71.0)	81.4 (71.0, 88.6)	75.4 (59.3, 86.6)	67.1 (56.8, 76.0)	89.5 (78.1, 95.3)	76.9 (52.2, 91.0)	58.2 (26.5, 84.3)	70.8 (53.5, 83.6)	52.4 (30.3, 73.6)	45.8 (26.7, 66.2)
Center	70.4 (55.9, 81.7)	69.2 (55.7, 80.0)	39.1 (26.9, 53.0)	85.0 (79.2,89.4)	64.7 (55.3, 73.0)	65.8 (57.1, 73.6)	81.2 (72.2, 87.7)	**47.9 (36.4, 59.5)**	67.0 (55.2, 77.0)	71.3 (62.2, 78.9)	68.4 (55.4, 79.0)	47.2 (37.0, 57.7)
Mexico City and State of Mexico	78.3 (63.1, 88.4)	70.2 (49.5, 85.0)	43.2 (26.0, 62.1)	88.5 (76.1, 94.9)	70.5 (56.9, 81.3)	70.5 (59.0, 79.8)	**72.3 (56.8, 83.9)**	67.9 (55.4, 78.2)	75.3 (57.6, 87.3)	58.8 (44.7, 71.7)	77.0 (64.5, 86.1)	60.3 (46.5, 72.6)
South	75.4 (65.6, 83.1)	81.1 (71.5, 88.0)	**35.2 (26.1, 45.6)**	77.2 (68.9, 83.8)	63.3 (53.4, 72.2)	**49.3 (40.8, 57.7)**	80.9 (73.5, 86.5)	53.2 (41.1, 64.9)	54.5 (42.7, 65.7)	55.3 (42.0, 67.8)	66.8 (52.3, 78.6)	38.4 (27.3, 50.8)
**Socioeconomic status**												
Low	70.5 (60.5, 78.9)	82.0 (71.2, 89.4)	38.0 (28.2, 49.0)	75.7 (68.8, 81.5)	71.2 (61.7, 79.2)	58.8 (50.7, 66.5)	75.2 (67.1, 81.9)	61.7 (51.3, 71.2)	58.1 (49.2, 66.6)	49.5 (35.6, 63.5)	64.4 (50.8,76.0)	37.3 (26.3, 49.9)
Medium	71.4 (59.0, 81.3)	74.3 (62.4, 83.5)	39.2 (27.3, 52.5)	81.0 (73.9, 86.6)	72.0 (64.4, 78.5)	59.2 (51.1, 66.9)	83.7 (75.7, 89.5)	52.2 (38.6, 65.5)	63.2 (48.3, 76.0)	64.1 (53.0, 73.8)	63.4 (48.0, 76.5)	48.5 (35.8, 61.4)
High	78.7 (66.1, 87.5)	**65.4 (51.7, 76.9)**	47.0 (34.2, 60.1)	**87.2 (79.9, 92.1)**	60.4 (49.9, 70.0)	64.2 (55.6, 72.0)	83.1 (73.0, 89.9)	60.6 (45.2, 74.1)	64.1 (44.4, 80.0)	**70.6 (60.7, 78.9)**	67.2 (52.0, 79.5)	47.9 (36.8, 59.2)
**Body mass index**^a^ (kg/m^2^)												
Normal	72.3 (65.1, 78.5)	74.9 (67.0, 81.4)	41.8 (34.3, 49.7)	78.5 (72.8, 83.3)	76.5 (70.8, 81.4)	59.1 (53.2, 64.8)	68.1 (61.0, 74.5)	69.0 (62.8, 74.5)	60.3 (52.3, 67.8)	60.8 (47.6, 72.6)	70.8 (57.1,81.6)	45.4 (33.7, 57.8)
Overweight	71.2 (29.6, 93.6)	**97.9 (90.3, 99.6)**	41.6 (14.0, 75.7)	84.7 (75.4, 90.9)	**60.3 (48.7, 70.8)**	60.2 (48.4, 70.9)	**82.4 (75.8, 87.5)**	75.7 (65.5, 83.6)	70.1 (60.6, 78.1)	61.7 (51.5, 71.0)	73.0 (59.7, 83.1)	47.9 (37.9, 58.1)
Obesity	100	40.3 (11.0, 78.8)	68.4 (28.8, 92.1)	**88.0 (80.0, 93.1)**	66.6 (49.4, 80.2)	61.5 (48.1, 73.3)	**86.8 (72.7, 94.2)**	81.5 (58.1, 93.3)	50.6 (19.4, 81.3)	68.0 (56.7, 77.4)	56.4 (41.7, 70.0)	44.0 (32.0, 56.8)

*n*, sample size; N, expanded sample. **Bold** numbers mean statistically significant difference vs reference category (p < 0.05), using logistic regression models. The reference category is the first row of each variable; except for the first variable (age), which is the in first column (preschool children). Data present percentages and confidence intervals (95%) and were adjusted by the survey design. ^a^ Body mass index (BMI): <25 kg/m^2^ (normal); 25–29.9 kg/m^2^ (overweight); and ≥30 kg/m^2^ (obesity). For those under 19 years of age, BMI for age was used according to the WHO growth standards. ^1^ We used the references for adequate intake (AI) by age and sex for sodium and potassium for those ≤18 years from the National Academies of Sciences, Engineering, and Medicine 2019. Dietary Reference Intakes for Sodium and Potassium. Washington, DC: The National Academies Press. For those >18 years, the references for sodium and potassium intake used came from WHO. Guideline: Potassium intake for adults and children. Geneva, World Health Organization (WHO), 2012 and WHO. Guideline: Sodium intake for adults and children. Geneva, World Health Organization (WHO), 2012. ^2^ High sodium was determined when sodium intake was above 800 mg (1–3 years old), 1000 mg (4–8 years old), 1200 mg (9–13 years old), 1500 mg (14–18 years old), and 2000 mg (19 years old or older). Insufficient potassium was considered when potassium intake was under 2000 mg (1–3 years old), 2300 mg (4–8 years old), 2500 mg (males 9–13 years old), 2300 mg (females 9–13 years old), 3000 mg (males 14–18 years old), and 3510 mg (19 years old or older). ^3^ Na-K: Those that exceed the ratio of >1.0 were considered.

**Table 4 nutrients-14-00281-t004:** Nutrition status and clinical characteristics according to quartiles of sodium and potassium intake in adults: ENSANUT-2016.

**Sodium Intake Quartiles (mg/day)**	** *n* **	**First**		**Second**		**Third**		**Fourth**		** *p* **
		**Mean**	**95%CI**	**Mean**	**95%CI**	**Mean**	**95%CI**	**Mean**	**95%CI**	
	**1356**	**801.8**	**(761.9, 841.7)**	**1578.93**	**(1517.6, 1649.2)**	**2438.4**	**(2381.6, 2494.9)**	**5049.6**	**(4579.4, 5519.8)**	**-**
**BMI (kg/m^2^) ^a^**	1294	28.7	(27.2, 30.2)	28.5	(27.2, 29.8)	28.7	(27.0, 30.3)	28.2	(27.1, 29.4)	0.966
**Waist circumference (cm).** Women	554	96.4	(91.4, 101.3)	93.5	(88.9, 98.2)	94.2	(90.2, 98.2)	94.1	(90.8, 97.4)	0.853
Men	356	98.562	(89.8, 107.3)	95.0	(90.9, 99.1)	98.1	(93.8, 102.5)	97.3753	(92.0, 102.8)	0.749
**SBP (mm Hg)**	901	123.2	(119.7, 126.8)	128.3	(119.6, 136.9)	120.5	(117.1, 123.9)	122.1	(117.2, 127.0)	0.373
**DBP (mm Hg)**	901	73.7	(71.3, 76.2)	76.4	(73.1, 79.7)	73.4	(70.9, 75.8)	72.1	(69.6, 74.6)	0.259
**eGFR (mL/min/1.73 m^2^)**	922	116.6	(110.6, 122.5)	113.7	(107.4, 120.1)	123.3	(108.2, 138.4)	112.4	(108.3, 116.5)	0.486
**Glucose (mg/dL)**	922	107.4	(101.4, 113.4)	115.8	(96.0, 135.5)	109.9	(98.0, 121.7)	101.0	(95.9, 106.0)	0.185
**Cholesterol (mg/dL) ^b^**	922	202.5	(189.0, 216.0)	193.5	(178.4, 208.6)	184.4	(172.8, 196.1)	**181.4**	**(173.6, 189.1)**	0.058
**HDL-c (mg/dL) ^c^**	922	40.8	(38.1, 43.6)	**35.7**	**(31.6 39.8)**	38.9	(35.2, 42.6)	**35.5**	**(33.2, 37.8)**	**0.026**
**LDL-c (mg/dL) ^d^**	865	123.6	(110.1, 137)	115.3	(108.3, 122.3)	111.9	(102.3, 121.4)	**106.6**	**(98.8, 114.3)**	0.156
**TG (mg/dL), mean (95%CI) ^e^**	922	197.8	(176.3, 219.3)	361.5	(79.9, 643.1)	218.8	(153.4, 284.3)	220.5	(170.3, 270.6)	0.547
**Diabetes ^f^ (%)**										
Prediabetes	926	24.6	(17.4, 33.6)	27.79	(15.0, 45.6)	32.92	(21.7, 46.6)	28.46	(16.5, 44.6)	0.548
Previous diagnosis		12.31	(7.9, 18.8)	8.107	(4.6, 13.8)	7.305	(4.2, 12.4)	6.175	(3.3, 11.2)	
Survey finding		9.9	(3.7, 24.2)	15.3	(3.7, 46.4)	9.1	(2.4, 28.6)	3.7	(1.3, 9.9)	
**High blood pressure ^g^ (%)**										
Normal	901	43.6	(31.3, 56.8)	32.0	(20.3, 46.5)	42.3	(29.7, 56.0)	46.8	(33.2, 60.9)	0.213
Elevated		11.5	(6.7, 19.2)	11.8	(4.9, 25.7)	18.3	(8.6, 34.9)	15.3	(8.4, 26.4)	
Stage 1		23.7	(13.3, 38.6)	20.8	(9.6, 39.4)	21.7	(11.1, 38.0)	9.8	(5.5, 17.0)	
Stage 2		7.9	(4.5, 13.5)	25.6	(10.4, 50.6)	8.4	(4.6, 15.0)	11.4	(4.4, 26.5)	
Previous diagnosis		13.2	(8.7, 19.6)	9.7	(5.6, 16.3)	9.3	(4.5, 18.3)	16.7	(8.0, 31.6)	
**Coronary heart disease (%)**	926	0.7	(0.1, 3.4)	1.5	(0.6, 4.2)	1.9	(0.6, 5.7)	4.1	(0.7, 21.7)	0.385
**Cerebro-vascular disease (%)**	916	0.0		0.5	(0.1, 3.3)	0.1	(0.0, 1.1)	0.9	(0.1, 6.0)	0.580
**Smoking (%)**										
Never	926	43.2	(32.0, 55.1)	65.8	(51.5, 77.8)	54.9	(40.8, 68.2)	31.8	(21.3, 44.5)	**0.002**
Current smoker		20.1	(11.4, 33.1)	6.4	(3.6, 11.1)	13.7	(7.0, 25.2)	30.9	(18.4, 46.9)	
Ex-smoker		36.7	(26.2, 48.7)	27.8	(17.4, 41.3)	31.4	(20.2, 45.3)	37.3	(24.8, 51.8)	
**Potassium intake quartiles (mg/day)**	** *n* **	**First**		**Second**		**Third**		**Fourth**		** *p* **
		**Mean**	**95%CI**	**Mean**	**95%CI**	**Mean**	**95%CI**	**Mean**	**95%CI**	
	**1356**	**965.4**	**(921.1, 1009.7)**	**1552.2**	**(1510.6, 1593.7)**	**2210.9**	**(2166.5, 2255.5)**	**5039.5**	**(4338.9, 5740.1)**	**-**
**BMI (kg/m^2^) ^a^**	1294	28.3	(27.1, 29.5)	27.4	(26.4, 28.5)	29.4	(28.2, 30.6)	28.6	(27.3, 29.9)	0.096
**Waist circumference (cm).** Women	554	98.2	(93.5, 102.9)	93.6	(89.4, 97.7)	96.6	(93.0, 100.3)	**92.1**	**(88.9, 95.4)**	0.143
Men	356	91.4	(87.5, 95.3)	95.7	(91.3, 100.0)	**98.4**	**(94.8, 102.1)**	**98.2**	**(93.4, 103.1)**	0.056
**SBP (mm Hg)**	901	122.0	(117.8, 126.2)	122.4	(116.8, 127.9)	122.3	(119.6, 125.0)	123.7	(118.6, 128.8)	0.959
**DBP (mm Hg)**	901	73.9	(70.6, 77.2)	74.5	(71.7, 77.2)	74.4	(72.1, 76.7)	72.8	(70.5, 75.2)	0.758
**eGFR (mL/min/1.73 m^2^)**	922	115.2	(112.0, 118.4)	117.6	(110.4, 124.7)	118.0	(113.1, 122.8)	115.3	(107.2, 123.4)	0.819
**Glucose (mg/dL)**	922	118.0	(98.0, 138.0)	102.0	(97.1, 106.9)	105.9	(99.2, 112.6)	106.7	(97.5, 115.9)	0.289
**Cholesterol (mg/dL) ^b^**	922	181.8	(172.1, 191.6)	187.9	(176.9, 198.8)	**198.7**	**(187.8, 209.5)**	184.1	(174.7, 193.6)	0.112
**HDL-c (mg/dL), ^c^**	922	41.6	(39.3, 43.9)	39.9	(33.3, 46.5)	40.1	(37.7, 42.4)	**34.3**	**(32.4, 36.1)**	**0.000**
**LDL-c (mg/dL) ^d^**	865	104.9	(95.7, 114.1)	112.7	(104.2, 121.2)	**123.5**	**(113.5, 133.5)**	108.7	(101.4, 115.9)	**0.036**
**TG (mg/dL), mean (95%CI) ^e^**	922	225.6	(145.9, 305.2)	224.9	(113.0, 336.9)	235.3	(160.9, 309.8)	258.7	(145.3, 372.1)	0.969
**Diabetes ^f^ (%)**										
Prediabetes	926	24.7	(14.8, 38.3)	29.3	(16.8, 46.0)	24.8	(14.9, 38.4)	31.5	(20.4, 45.2)	0.505
Previous diagnosis		9.2	(5.3, 15.5)	9.8	(5.8, 16.0)	12.3	(7.5, 19.5)	5.0	(2.8, 8.9)	
Survey finding		17.8	(4.5, 50.0)	5.4	(2.5, 11.4)	5.2	(1.5, 16.9)	8.0	(2.6, 22.2)	
**High blood pressure ^g^ (%)**										
Normal	901	47.4	(31.5, 63.9)	41.3	(28.0, 56.2)	36.1	(25.8, 47.9)	43.4	(31.2, 56.4)	0.754
Elevated		10.1	(4.9, 19.6)	9.4	(5.0, 16.8)	20.3	(10.4, 35.8)	15.9	(8.7, 27.5)	
Stage 1		24.5	(9.3, 50.6)	22.9	(11.8, 39.8)	19.3	(10.6, 32.4)	12.7	(7.6, 20.6)	
Stage 2		8.9	(4.2, 18.1)	12.7	(5.1, 28.2)	11.0	(5.9, 19.5)	14.8	(6.5, 30.1)	
Previous diagnosis		9.1	(5.1, 15.6)	13.7	(8.1, 22.3)	13.3	(7.9, 21.6)	13.2	(6.3, 25.6)	
**Coronary heart disease (%)**	926	1.0	(0.3, 3.5)	0.7	(0.1, 4.6)	3.2	(1.2, 7.9)	3.2	(0.6, 16.8)	0.496
**Cerebro-vascular disease (%)**	916	0.7	(0.1, 5.3)	0.0	0	0.2	(0.0, 1.6)	0.7	(0.1, 4.6)	0.699
**Smoking (%)**	926									
Never		58.2	(42.3, 72.5)	50.8	(36.7, 64.7)	45.4	(33.7, 57.7)	42.5	(30.7, 55.2)	0.691
Current smoker		17.8	(8.9, 32.2)	20.2	(10.8, 34.5)	22.4	(12.9, 36.0)	19.6	(10.3, 34.0)	
Ex-smoker		24.1	(15.5, 35.4)	29.1	(19.6, 40.8)	32.2	(22.1, 44.2)	37.9	(26.7, 50.7)	

Data adjusted by survey design. **Bold** numbers indicate a statistically significant difference between first quartile and other quartiles using Wald test for continuous variables (*p* < 0.05). ^a^ Body mass index (BMI): <25 kg/m^2^ (normal); 25–29.9 kg/m^2^ (overweight); ≥30 kg/m^2^ (obesity). ^b^ High total cholesterol levels: ≥200 mg/dL. ^c^ Low HDL-c levels (hypoalphalipoproteinemia): <40 mg/dL. ^d^ High LDL-c levels: ≥100 mg/dL. ^e^ High triglycerides levels: ≥150 mg/dL. ^f^ Diabetes classification: prediabetes (fasting glucose ≥ 100 y < 126 mg/dL or HbA1c ≥ 5.7 and < 6.5%); survey finding (fasting glucose ≥ 126 mg/dL or HbA1c ≥ 6.5%). ^g^ Blood pressure (mm Hg): normal (<120/80); elevated (systolic between 120–129 and diastolic < 80); stage 1 (systolic between 130–139 or diastolic between 80–89); stage 2 (systolic at least 140 or diastolic at least 90.

## Data Availability

The data that support the findings of this study are available from the correspondence author upon reasonable request.

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
