# Peer review of "Dietary Sodium and Potassium Intake: Data from the Mexican National Health and Nutrition Survey 2016"

_nutrients, 2022, doi:10.3390/nu14020281_

Round 1

Reviewer 1 Report

This is a straightforward epidemiological study.

I believe this is correctely planned and conducted.

More important, I agree with the main conclusions, where the high sodium diet risk seems well underlined.

Some pitfalls however appear:

Table 1: in the caption no definition are offered to describe the different meaning of "n" and "N". This should be made clear.

rows 417-422: some mistakes produced a sort of confusing repetition.

row 655: may occur be because... "be" should be erased.

row 700: "potassium" is repeated twice.

I can't find throughout text and tables (and even supplementary material) where data about urinary spot sodium and potassium measurement appear.

The disclosure about the limits and the insufficiency of dietary recall methods in the identification of sodium and potassium consumption should be amplified and stressed.

Author Response

Dear reviewer,We appreciate your time and availability to review this draft. Also appreciate your comments and suggestions, which we have addressed each of them. In the attached document you will find each of them highlighted in yellow, and in the following text you will find the answer to each of the observations made.Point (P) 1: Table 1: in the caption no definition are offered to describe the different meaning of "n" and "N". This should be made clear.

Response (R) 1: The meaning of "n" and "N" was placed in the subtitle of table 1 and table 3 where these acronyms are used.

P2: rows 417-422: some mistakes produced a sort of confusing repetition.R2: Thanks for the observation, due to the confusion generated by the marked lines it was rewritten for better understanding. Lines 409-411.P3: row 655: may occur be because... "be" should be erased.R3: Thanks for the detailed observation, the word was removed. Lines 353-355P4: row 700: "potassium" is repeated twice.R4: Thanks for the detailed observation, the duplicate word was removed. Lines 397-398P5: I can't find throughout text and tables (and even supplementary material) where data about urinary spot sodium and potassium measurement appear.R5: Thanks for your observation, sodium and potassium intake was only estimated from 24-hour food recalls, however, we understand the confusion due to it being described in urine biomarkers. Therefore, the paragraphs of serum and urine biomarkers were described to understand the classifications in a better way. Lines 147-156P6: The disclosure about the limits and the insufficiency of dietary recall methods in the identification of sodium and potassium consumption should be amplified and stressed.R6: Thanks for the recommendation, the strengths and limitations of the study have been edited. The limitations of the method used to estimate sodium and potassium intake were increased. Lines 435-358

We will be attentive to any additional changes

Regards

Reviewer 2 Report

See a few comments in the attached pdf file.

Author Response

Dear reviewer,We appreciate your time and availability to review this draft. Also appreciate your comments, which we have attended to each of them. In the attached document you will find each of them highlighted in yellow. Here are some relevant changes to the text in which the changes were made are highlighted:Point 1:  AbstractLine 17, we change "...in 3132 adults" to "...3132 in adults"Point 2: IntroductionLine 39, we rewrite the confused sentence.Line 40, we add "study" after the word "cohort"Point 3: MethodsLines 147-156, the paragraphs of serum and urine biomarkers were described to understand the classifications in a better way.Line 149, the word "y" was changed to the word "and"Line 158, the word "was" was changed to the word "where"Point 4: ResultsLine 243, the word "an" was changed to the word "a"Point 5: DiscussionLines 353-355, in the entence the word "be" was removed.Lines 435-358, the strengths and limitations of the study have been edited.

We will be attentive to any additional changes

Regards

Round 2

Reviewer 1 Report

I believe that my observations have now been thoroughly fullfilled adn the paper is now acceptable for publication on a high-impact journal.

This manuscript is a resubmission of an earlier submission. The following is a list of the peer review reports and author responses from that submission.

Round 1

Reviewer 1 Report

This papers addresses a critically important topic, but some pitfalls decrease the interest in such paper.

First and foremost, the authors employed an insufficient methodology to evaluate sodium consumption (anamnestic records cannot substitute for 24h sodium excretion).

Moreover they do not acknowledge this major limit of their study in a strong way.

Furthermore, many other issues appear in the text:

row 97 - "all males > 1 years old" (?)

row 140 - height is never measured in squared meters (!), instead a squared height is employed in the calculations of BMI

row 151 - there is an updated version of ref. 32

Results section

row 203 - how the representativity of the sample studied has been calculated?

row 225 - "fruits (2798 mg)..." how is it possible?

Table 4 - the third sodium quartile shows the lowest systolic BP and the highest prevalence of hypertension. How is this possible?

row 335 - the cited review is not by GRADUAL et al., but by GRAUDAL et al.

row 365 - I can't really figure how vegetables and fruit can contribute substantially to sodium consumption.